# Newly Developed Pediatric Membrane Oxygenator that Suppresses Excessive Pressure Drop in Cardiopulmonary Bypass and Extracorporeal Membrane Oxygenation (ECMO)

**DOI:** 10.3390/membranes10110362

**Published:** 2020-11-21

**Authors:** Makoto Fukuda, Asako Tokumine, Kyohei Noda, Kiyotaka Sakai

**Affiliations:** 1Department of Biomedical Engineering, Kindai University, 930 Nishimitani, Kinokawa-city, Wakayama 649-6493, Japan; tokumine@waka.kindai.ac.jp; 2Department of Medical Engineering, Siga University of Medical Science Hospital, Seta, Tsukinowa-cho, Otsu-City, Siga 520-2192, Japan; noda0611@icloud.com; 3Department of Chemical Engineering, Waseda University, 3-4-1 Okubo, Shinjuku-ku, Tokyo 169-8555, Japan; kisakai@waseda.jp

**Keywords:** membrane oxygenator, hollow fiber membrane, cardiopulmonary bypass (CPB), extracorporeal membrane oxygenation (ECMO), COVID-19

## Abstract

This article developes a pediatric membrane oxygenator that is compact, high performance, and highly safe. This novel experimental approach, which imaging the inside of a membrane oxygenator during fluid perfusion using high-power X-ray CT, identifies air and blood retention in the local part of a membrane oxygenator. The cause of excessive pressure drop in a membrane oxygenator, which has been the most serious dysfunction in cardiovascular surgery and extracorporeal membrane oxygenation (ECMO), is the local retention of blood and air inside the oxygenator. Our designed blood flow channel for a membrane oxygenator has a circular channel and minimizes the boundary between laminated parts. The pressure drop in the blood flow channel is reduced, and the maximum gas transfer rates are increased by using this pediatric membrane oxygenator, as compared with the conventional oxygenator. Furthermore, it would be possible to reduce the incidents, which have occurred clinically, due to excessive pressure drop in the blood flow channel of the membrane oxygenator. The membrane oxygenator is said to be the “last stronghold” for patients with COVID-19 receiving ECMO treatment. Accordingly, the specification of our prototype is promising for low weight and pediatric patients.

## 1. Introduction

Membrane oxygenator is the artificial organ (artificial lung) used in cardiopulmonary bypass (CPB), such as coronary artery bypass surgery, valve replacement surgery, implantable artificial heart surgery. Additionally, it is used in extracorporeal membrane oxygenation (ECMO) for respiratory assistance therapy to acute respiratory distress syndrome, such as that caused by new coronavirus infection (2019 novel coronavirus disease, COVID-19) [1,2,3]. Membrane oxygenator is said to be the “last stronghold” for patients with COVID-19 [1].

The most serious dysfunction (adverse event) of a membrane oxygenator is abnormally increased pressure drop across oxygenating and heat exchanging sections of a membrane oxygenator. Platelet-fibrin thrombus and coagulation in membrane oxygenator block blood flow channel and increase pressure drops, making it difficult to perfuse blood into the membrane oxygenator with a systemic centrifugal pump despite adequate anticoagulation [4,5,6,7,8]. In this case, it is necessary to perform an emergency oxygenator changeover that is originally used with another unused membrane oxygenator. During this replacement, CPB and ECMO treatment are discontinued. It is a serious incident that puts the patient’s life endanger. To reduce such incidents and accidents, it is necessary to detect the occurrence of excessive pressure drops across membrane oxygenator in advance, and the pressure drops and the blood flow channel pressures at the inlet and outlet of the membrane oxygenators (1st pressure, 2nd pressure) are recommended to be monitored [5,6,7,8]. Fisher et al. reported that the incidence of excessive pressure drop was 0.4–2.5% [6]. On the other hand, the Working group of the Japanese Society of Cardiovascular Surgery in Japan found that the incidence of membrane oxygenator replacement, due to excessive pressure drop was 1/3578 (0.03%) in adults and 1/576 (0.17%) in children (2016), and reported that there were six times more incidents of membrane oxygenator replacement in children than those in adults [9,10].

Although many membrane oxygenators for adults with completely different designs have been developed for practical use in Japan, there are very few pediatric membrane oxygenators. Since the frequency of membrane oxygenator replacement was six times higher in children than in adults, it was a problem from the viewpoint of medical safety that there were not many options for pediatric membrane oxygenators. When a conventional membrane oxygenator for adults is used for a low weight patient, the blood velocity is slow because of its large priming volume, so that the gas transfer rate is low. Additionally, the actual retention time is long so that blood coagulation occurs and blocks the blood flow channel. Therefore, a properly designed small oxygenator is urgently needed for low weight or pediatric patients.

Numerous studies have been conducted on the design of membrane oxygenators [11,12,13,14,15,16,17,18]. Matsuda et al. clarified the relationship between the appropriate shape of hollow fiber bundle and gas transfer rate of an outside blood flow membrane oxygenator by an experimental approach from a transport phenomenon perspective [12,13]. On the other hand, a lot of knowledge has been obtained by Computational Fluid Dynamics (CFD) [14,16,17,18], Zhang et al. developed the design methodology to optimize a design of membrane oxygenator by calculating blood velocity and pressure drop distributions in local areas inside a blood flow path using the CFD simulation [16]. However, incidents are still occurring, due to excessive pressure drop. As a matter of fact, it has not been clear what kind of phenomenon or membrane oxygenator design cause an excessive pressure drop, and it has not been solved yet [10,11]. CFD simulation (mathematical model) is useful, but it has limitations in grasping the actual phenomena. Therefore, the problem is solved by understanding the actual phenomena through an experimental approach. Here, we focused on the local retention of blood and air inside of a membrane oxygenator as one of the causes of the excessive pressure drop, and developed an original experimental method using high-power X-ray computed tomography (CT) for verification. Needless to say, platelet-fibrin thrombus and coagulation in membrane oxygenator are not only caused by hydrodynamic reasons, but also materials and surface roughness.

The objective of the present study is to perform internal imaging of the membrane oxygenator during fluid perfusion using a high-power X-ray CT device, and to locate the local blood and air retention. To reduce the incidents, due to excessive pressure drop in cardiovascular surgery and ECMO, we propose the design concept of a membrane oxygenator that suppress such blood retention and excessive pressure drop. In addition, we aim to make a prototype and evaluate the pediatric membrane oxygenator.

## 2. Material and Methods

### 2.1. Hollow Fiber Membrane Oxygenators

The membrane oxygenators used in this study were commercial membrane oxygenators; oxia^®^ ACF (JMS Co., Ltd., Tokyo, Japan, Sample A), CAPIOX^®^ FX05 (Terumo Co., Ltd., Tokyo, Japan, Sample C), and the newly developed pediatric membrane oxygenator (prototype), and commercial available pediatric membrane oxygenator, which is currently the smallest in the world (LivaNova Co., Ltd., Sorin Group, Mirandola, Italy, Sample B) as a control. Table 1 shows the specifications of the hollow fiber membrane oxygenators. These are the outside blood flow membrane oxygenators.

The membrane specifications of the new pediatric oxygenator and Sample A are exactly the same. The gas permeabilities of the prototype and Sample B membranes are almost the same.

Calculated or ideal retention time is the time that blood is ideally or uniformly perfused once inside a device. The actual retention time is different from the calculated retention time.

### 2.2. Imaging a Membrane Oxygenator Inside During Fluid Perfusion Using High-Power X-Ray CT

We have developed the non-destructive flow analysis experiment and evaluated the retention of air in the blood channel of membrane oxygenators [20]. When fluid flows inside a membrane oxygenator composed of the metal heat exchanger, gas exchanger, and others made from various materials is investigated using a medical X-ray CT (tube voltage output, 120 kV), artifacts are generated, false images and obstructions occur during imaging, so accurate imaging is impossible. Therefore, high-power (430 kV) X-ray CT was used for high-resolution imaging of membrane oxygenators composed of various materials, such as stainless-steel and polypropylene hollow fiber membranes during fluid perfusion. The resolution is significantly higher than that of medical X-ray (120 kV). Figure 1 shows the photograph of the experimental apparatus. Figure 2 shows the prototype of a closed extracorporeal circuit. 

We briefly show the experimental method based on the patent already reported [20]. A closed-circuit system including membrane oxygenator, centrifugal pump, etc., were installed in the 2-stage set of the frame, and only a membrane oxygenator was imaged.

An extracorporeal circuit with a small priming volume was created by hand so that a required flow rate (0.5–8 L/min) could be perfused with a small fluid volume, and it was connected to a soft reservoir for sealing the circuit (closed-circuit, Figure 2). The centrifugal pump was connected to the controller. Next, reverse osmosis (RO) water was injected into a soft reservoir, membrane oxygenator, and a circuit to perform priming (priming volume ≈ 700 mL). After that, RO water and an X-ray contrast agent (RO aqueous solution of 10 wt% BaSO_4_, BaSO_4_ particle size 5 µm, specific gravity 4.3 g/cm^3^) were perfused at an average flow rate of 5 L/min (0.7 L/min for a pediatric membrane oxygenator) for 60 min. The flow was in a steady-state, during which X-ray imaging was performed. RO aqueous solution of 10 wt% BaSO_4_ (BaSO_4_ particle size 5 µm, specific gravity 4.3 g/cm^3^) was used as an X-ray contrast agent for the following reasons.
(1)A test solution that can be used for X-ray CT imaging (high specific gravity of particles).(2)It can localize in the region of the blood flow channel where it is expected to stay easily in between the hollow fibers (the red blood cell diameter is 8 µm as a guide, and since the BaSO_4_ particles do not deform during fluid perfusion, 5 µm particle was used as the BaSO_4_ particle size which is slightly smaller than 8 µm).(3)On the other hand, during the X-ray CT imaging, there should be no flow channel blockage that would prevent the test solution from flowing.

The tests were carried out on several copies of the prototype. They were not reused.

### 2.3. High-Power X-Ray Computed Tomography (CT)

TOSCANER-24500twin (Toshiba Co., Ltd., Tokyo Japan, installed at the Industrial Technology Center of Wakayama Prefecture) was used as the High-power (430 kV) X-ray computed tomography (CT) equipment. The high-power X-ray computed tomography (CT) equipment was operated in compliance with the Regulation on Prevention of Ionizing Radiation Hazards [21]. The area around the high-power X-ray CT system is outside the controlled area with an effective dose of 2 μSv/h or less (≈1.3 mSv/every three months or less). The operation of the high-power X-ray CT system was performed by a certified Operation Chief of Work with X-rays.

The imaging area obtained by the detector was decomposed with a pixel size of 2048 × 2048 to create voxels. Voxels change the color of CT images depending on the proportion of substances in a unit volume. When the substance density is high, it is displayed in white, and when there is a lot of air, it is displayed in black. The plane cut view image from the top was taken every 2 mm of height (thickness) of the devices.

### 2.4. Oxygen and Carbon Dioxide Transfer Rates, Pressure Drop of Blood Flow Channel

Fresh bovine blood containing heparin Na was adjusted to standard venous blood characteristics (Hb = 12.0 g/dL, Base Excess (BE) = 0 mmEq/dL, 37 ℃, P_V_CO_2_ = 47 ± 1 mmHg) [21]. This was perfused once through a membrane oxygenator to evaluate gas transfer rate. The ratio of blood flow rate to gas flow rate (V/Q) was V/Q = 0.5, 1, 2, and oxygen fractional concentration (FiO_2_) of inspired gas was 100%. Bovine blood was sampled at the blood inlet and outlet of the membrane oxygenator, and pH, PO_2_, PCO_2_, and oxygen saturation were measured with the blood gas analyzer (Rapid Lab 348EX, Siemens Healthcare Diagnostics Co. Ltd., Vienna, Austria). The oxygen contents of arterial and venous blood were calculated, and the oxygen and carbon dioxide transfer rates were calculated based on Fick’s equation (law of conservation of mass) (*n* = 3) [22,23],
(1)$$K_{O2}=\left(\frac{S_{a}O_{2}-S_{v}O_{2}}{100}\times 1.34\times Hb+0.00314\times \left(P_{a}O_{2}-P_{v}O_{2}\right)\right)\times \frac{Q_{B}}{100}$$
(2)$$K_{CO2}=\left(\frac{C_{v}CO_{2}-C_{a}CO_{2}}{100}\right)\times \frac{Q_{B}}{100}=\left(\frac{2.226\times T_{v}CO_{2}-T_{a}CO_{2}}{100}\right)\times \frac{Q_{B}}{100}$$
where *K_O2_* is the oxygen transfer rate (mL/min), *SaO_2_* the arterial saturation (%), *SvO_2_* the venous saturation (%), *PaO_2_* the arterial partial pressure (mmHg), *PvO_2_* the venous partial pressure, *Hb* the hemoglobin concentration (g/dL), *Q_B_* the flow rate of blood side (mL/min), 1.34 the Hemoglobin oxygen capacity (mL/g), 0.00314 the oxygen solubility (vol%/mmHg, 37 °C), *K_CO2_* the carbon dioxide transfer rate (mL/min), *CaCO_2_* the arterial carbon dioxide content (mL/dL), *CvCO_2_* the venous carbon dioxide content (mL/dL), *TaCO_2_* the arterial carbon dioxide content (mmol/dL), *TvCO_2_* the venous carbon dioxide content (mmol/dL), 2.226 the conversion factor ((mmol/L)→(mL/dL), 1 mol = 22.26L at CO_2_).

The blood inlet pressure P_Bi_ and the outlet pressure P_Bo_ of a membrane oxygenator were measured with the digital pressure sensor PPX-R01N-6M (CKD CORPORATION Ltd., Tokyo, Japan) at the blood flow rate Q_B_ = 0.1–2.0 L/min. The pressure drop ΔP_B_ in the blood flow channel was calculated.
(3)$$\Delta P_{B}=P_{Bi}-P_{Bo}$$

Data on the membrane oxygenators were treated using a two-way analysis of variance (two-way ANOVA, *p* < 0.05).

## 3. Results

### 3.1. Non-Destructive Visualization of a Hollow Fiber Membrane Oxygenator Using High-Power X-Ray Computed Tomography

Figure 3 shows the high-power X-ray CT images of Sample A. Figure 3a is the external view of the device, and Figure 3b is the vertical cross-sectional view. Figure 3c shows the photographic image of the vertical cross-sectional view. The heat exchangers appear white, and the packed hollow fiber membrane (gas exchanger) containing air is clearly observed, due to the difference in CT value (Hounsfield Unit) of the material and structure. The packed hollow fiber membrane layer containing air is arranged in the circular blood channel. The outer periphery of the blood channel is in a state where the adhesive has infiltrated into the hollow fibers and solidified, and it is distinguished from the circular blood channel.

Figure 4a shows the high-power X-ray CT image of the vertical cross-sectional view of Sample C, and Figure 4b shows the photographic image of the external view.

Figure 5 shows the plane cut views of the yellow arrow in the vertical view image of Figure 3b. Figure 5(b)-1 shows the interface between the heat exchanger and blood flow channel. A large number of stainless-steel tubes forming the heat exchanger are arranged in the lateral direction and appear to overlap the blood flow channel. Figure 5(b)-2 shows the plane cut view about 2 mm above Figure 5(b)-1. Figure 5(b)-2 is the interface between the heat exchanger and spacers, the air layer in between the spacers was observed to be black. Figure 5(b)-4 is the interface between the heat exchanger and the packed hollow fiber membrane. Figure 5(b)-5 is the packed hollow fiber membrane layer, and the area inside the yellow dotted circle is the blood flow channel. Figure 5(b)-6 is the interface between the packed hollow fiber membrane layer and the arterial filter. The wavy arterial filter was observed to be white. As described above, non-destructive visualization by high-power X-ray computed tomography (CT) enables detailed observation of the internal structure of the membrane oxygenator composed of various materials.

### 3.2. Non-Destructive Visualization in a Hollow Fiber Membrane Oxygenator during RO Water and X-Ray Contrast Agent Perfusion

Figure 6 shows the plane cut views of Sample A during RO water perfusion. There were air bubbles displayed in black at the interface between the spacers and the outer peripheral part (Figure 6(b)-2~4). The blood inlet is located in the bottom center of the device, the blood flow velocity is higher in the center and slower in the outer periphery. Therefore, uneven flow (channeling) and air/ blood retention are to occur at the outer periphery of the device. From Figure 6, air and blood retention occurs at the boundary surface and outer periphery of spacers constituting the laminated structure, which affects blood coagulation reactions and thrombosis.

Figure 7 shows the plane cut views of Sample A during RO aqueous solution of 10 wt% BaSO_4_ perfusion. X-ray contrast agent was displayed in white and air bubbles were displayed in black. Compared with Figure 6, retention of bubbles and X-ray contrast agent particles (BaSO_4_) were more noticeable in the heat exchanger, 1st, and 2nd spacers (Figure 7(b)-1~4). There were many air and BaSO_4_ particles at the interface between the spacers and the outer peripheral parts. From the results shown in Figure 7, there is a risk of excessive pressure drop, due to retention at the peripheral parts and the interfaces of the blood flow channel. This is consistent with the results of local blood coagulation inside a membrane oxygenator after clinical use. The local retention area of the fluid is grasped in more detail when the X-ray contrast agent (BaSO_4_ particles, specific gravity 4.3 g/cm^3^) is perfused than when RO is perfused. Therefore, the methodology in which the X-ray contrast agent is perfused is more useful as a critical test that reproduces a clinically serious situation, such as blood retention and coagulation.

On the other hand, for comparison, there were no X-ray contrast agent particles (BaSO_4_) in the blood flow channel of the gas exchanger (in between hollow fibers) (Figure 7(b)-5). It was feared that the hollow fibers would easily contact each other, and an effective gas transfer area would be smaller than 1.7 m^2^, but there was no such concern from Figure 7(b)-5. It was considered that the X-ray contrast agent was trapped in the arterial filter of Figure 7(b)-6 after passing through the gas exchanger (in between hollow fiber membranes) of Figure 7(b)-5. Based on these findings, we designed the new pediatric membrane oxygenator and made a prototype, as shown in Section 3.3. Although the quantitative analysis is not sufficient in these experiments, it is possible to clarify the local air and blood retention inside the membrane oxygenator, which has not been understood by the conventional CFD.

We have previously conducted similar experiments and found that it is difficult to detect the tracer in bovine blood with X-rays. Therefore, we tried to visualize the internal fluid flow inside a membrane oxygenator using only pure water and barium aqueous solution.

### 3.3. Design Concept of the Newly Developed Pediatric Membrane Oxygenator

Figure 8 shows the vertical cross-sectional photographic images of the newly developed pediatric membrane oxygenator (prototype) and commercially available pediatric membrane oxygenator device (Sample B). Figure 9 shows the three-dimensional diagram to explain the geometry of the prototype and the flow of gas and bloodstreams.

The pressure drop of a membrane oxygenator is affected by three factors, the blood characteristics of the patient (hematocrit value, viscosity, temperature) and blood flow rate, in addition to the complicated flow channel of a membrane oxygenator. During cardiopulmonary bypass (CPB), perfusionists change the blood flow rate and monitor the inlet pressure and outlet pressures of a membrane oxygenator, but cannot control the blood properties (hematocrit, viscosity, temperature). Therefore, even if the risk of patient factors cannot be controlled, it is desired to design a membrane oxygenator that allows perfusionists to flexibly handle the operation conditions. For this purpose, it is necessary to design a compact, high performance and highly safe membrane oxygenator as compared with the conventional oxygenators.

The basic design of the prototype is the structure in which heat exchanger (stainless-steel tubes), gas exchanger (packed hollow fiber membrane), filters, and spacers that partition them are laminated. The reed screen sheets composed of hollow fiber membranes are alternately laminated to attain the multilayer structure of the hollow fiber bundle. Blood flows from the bottom center of the device and is perfused in between the stainless-steel tubes and then crossflows through packed hollow fiber membrane, which is defined as the outside blood flow membrane oxygenator (Figure 8a, Figure 9). The contacting pattern between the blood and the exchange fluid (gas, heat carrier) is the crossflow. In the crossflow, the fluid streams flow perpendicular to each other to achieve the turbulent flow of blood, reducing the boundary layer of blood side and increasing gas transfer rate. The outside blood flow membrane oxygenator also has the advantages of lower blood priming volume and blood pressure drop, which allow the wide range of blood flow rate at clinical operations, compared with the inside blood flow membrane oxygenator [12,13,24]. The design of the membrane device determines the performance and safety of the membrane oxygenator. The characteristics of the currently used membrane oxygenators have been shown in the previous study [24]. Compared with these membrane oxygenators, the membrane oxygenator produced in our study has key parameters, such as membrane area, blood flow rate, and priming volume, that are an order of magnitude smaller [24], so our membrane oxygenator has completely different characteristics compared with previous membrane oxygenators. Therefore, our study is extremely useful from a practical point of view.

On the other hand, since the blood flow channel of a membrane oxygenator is much more complicated than that of human blood vessels, uneven flow (channeling) and retention of blood are likely to occur in the blood flow channel of a membrane oxygenator [11,12,13,14,15,16,17,18]. Uneven flow and retention generate blood coagulation reaction, block parts of the blood flow channel, and reduce the blood flow channel area, thus increasing the pressure drop. Therefore, the blood flow channel of the membrane oxygenators has a simple circular shape to suppress uneven flow and retention of blood. This design concept suppresses an uneven flow and retention and makes the blood flow velocity uniform.

From the results shown in Section 3.2, we focused on the fact that there were many local air retentions at the boundaries of the laminated parts in the membrane oxygenator. The concrete design concept is indicated as follows, (1) Assuming that air and blood retention is one of the causes of excessive pressure drop across membrane oxygenator, we designed the blood flow channel that minimizes the boundary parts of the laminated parts to suppress this. To obtain a smooth blood flow (to prevent blood retentions), a circular channel having a continuous heat exchanger and gas exchanger is designed. (2) The size of the prototype is aimed to be smaller than that of the commercially available pediatric membrane oxygenator, which is currently the smallest in the world (Figure 8). On the other hand, (3) To minimize pressure drop, the membrane area is set to 0.39 m^2^, and the priming volume is set to 37 mL, which is larger than the membrane area (0.22 m^2^) and priming volume (31 mL) of conventional pediatric membrane oxygenator. Moreover, (4) The maximum blood flow rate is set to 2.0 L/min so that the flow rate operation range becomes wider and increase the perfusionists’ degree of freedom, and the blood retention time is shortened to suppress the blood coagulation reaction.

On the other hand, the concentric circle type (Figure 4) is the mainstream in the design of membrane oxygenators worldwide, and the existence of such a completely different design concept is intriguing. In terms of biofluid mechanics, this design can maintain the pressure drop suppression and high gas transfer rate. This is because the blood flow is temporarily stored in the blood reservoir, and dynamic pressure of blood is converted into static pressure so that the blood flows uniformly into the packed capillary membranes.

### 3.4. Non-Destructive Visualization of the Newly-Developed Membrane Oxygenator during Fluid Perfusion

Figure 10 shows the vertical cross-sectional images of the newly-developed membrane oxygenator during RO water and RO aqueous solution of 10 wt% BaSO_4_ perfusion. Figure 11 shows the plane cut views during RO water perfusion, which is indicated by the red arrow in Figure 10a.

There were no air bubbles compared with Figure 10, and it was found that RO water was uniformly perfused in local areas. Figure 11(a)-4 and 5 are hollow fiber membrane bundle, and the area inside the yellow dotted circle is the blood flow channel. RO water is uniformly perfused in between the hollow fibers in the circular channel. The hollow fiber membranes and the housing adhered to each other with an adhesive around the area outside the yellow dotted circle.

Figure 12 shows the plane cut views of Figure 10b during an aqueous solution of 10 wt% BaSO_4_ perfusion, which is indicated by the red arrow in Figure 10b. Compared with Figure 7, there were no air bubbles, and few X-ray contrast agent particles (BaSO_4_) were observed. This is because the new design minimizes the circular flow path and boundary parts. Thus, the risk of excessive pressure drop in the prototype is less than that in sample A.

However, the problem that a slight amount of X-ray contrast agent retention is observed at the outermost periphery of the blood flow channel in Figure 12(b)-1 is for future work.

### 3.5. Pressure Drop of Blood Flow Channel

Figure 13a shows the relationship between the blood flow rate and the pressure drop of both pediatric membrane oxygenators within the recommended range of blood flow rate at clinical practices. The pressure drop of the newly developed pediatric membrane oxygenator (prototype) was much smaller than that of the commercially available membrane oxygenator (Sample B). This is mainly because the priming volume of the prototype (37 mL) is larger than that of Sample B (31 mL) and the length of the blood flow channel is shorter. The pressure drop at the maximum blood flow rate was also smaller in the new membrane oxygenator (Table 2). Thus, a risk of excessive pressure drop in the prototype is less than that in Sample B.

On the other hand, Figure 13b shows the relationship between the blood flow rate and the pressure drop of the prototype and Sample A. Within the recommended range of blood flow rate, the pressure drop of the prototype was about the same as that of Sample A.

However, at the same blood flow rate, the pressure drop of the prototype is higher than that of Sample A. One of the reasons that pediatric oxygenators are replaced more frequently than adult membrane oxygenators in CPB [6,9,10] is that the pressure drop (design value) of a pediatric oxygenator is higher than that of an adult oxygenator at the same blood flow rate, which can be seen in Figure 13b. Moreover, excessive pressure drop leading to incidents is likely to occur when air and blood retention is longer, and/or due to uneven flow (channeling). So it is essential to appropriately use the membrane oxygenator depending on the set blood flow rate at clinical practices. In the future, we will confirm the incidence, due to the excessive pressure drop at clinical practices using this pediatric membrane oxygenator.

Since the pressure drop with respect to the blood flow rate increases quadratically, the blood flow is turbulent flow and contributes to the increase in gas transfer rate [12,13,14,15,16].

### 3.6. Determination of Oxygen and Carbon Dioxide Transfer Rates

Figure 14a shows the relationship between the blood flow rate and oxygen transfer rate of both pediatric membrane oxygenators. The oxygen transfer rates of both oxygenators were almost the same, and there was no significant difference. This is because that the new membrane oxygenator has a larger priming volume (37 mL > 31 mL) and the blood flow velocity is low, and the resistance in the blood-side diffusion boundary layer is larger during oxygen transfer, thereby decreasing oxygen transfer rate per membrane area [11,12,13]. Figure 14b shows the relationship between V/Q and oxygen transfer rate at Q_B_ = 0.7 L/min. There is no significant difference in the oxygen transfer rate between the two membrane oxygenators at V/Q = 0.5, 1, and 2. Additionally, the oxygen transfer rate does not increase even if the gas flow rate is increased, under which the blood flow rate is limited.

Figure 15a shows the relationship between the blood flow rate and the carbon dioxide transfer rate of both pediatric membrane oxygenators. The carbon dioxide transfer rate of the prototype was higher at Q_B_ > 0.7 L/min. This is because the prototype has a larger membrane area. Figure 15b shows the relationship between V/Q and carbon dioxide transfer rate at Q_B_ = 0.7 L/min. Unlike the oxygen transfer rate results, the higher the V/Q, the higher the carbon dioxide transfer rate. This is because the gas flow rate (flow velocity) is higher and the carbon dioxide partial pressure at the gas side outlet is lower, so the overall difference of carbon dioxide partial pressure between the blood side and the gas side (driving force) is larger, thereby the carbon dioxide transfer rate is increased.

Table 2 shows the numerical values extracted from Figure 13, Figure 14 and Figure 15. The gas transfer rates at each maximum blood flow rate were higher in the new membrane oxygenator. It was feared that the hollow fibers of the new membrane oxygenator would be easily contacted with each other, and the effective gas exchange area would be smaller, but there was no such concern.

## 4. Discussion

As mentioned above in Section 3.3, the pressure drop of a membrane oxygenator is affected by three factors, the blood characteristics of the patient (hematocrit value, viscosity, temperature) and blood flow rate, in addition to the complicated flow channel of the membrane oxygenator. During cardiopulmonary bypass (CPB), perfusionists change the blood flow rate and monitor the inlet pressure and outlet pressures of a membrane oxygenator, but cannot control the blood properties (hematocrit, viscosity, temperature). Therefore, even if the risk of patient factors cannot be controlled, it is desired to design a membrane oxygenator that allows perfusionists to flexibly handle the operation conditions.

Since the blood flow channel of a membrane oxygenator is much more complicated than that of human blood vessels, uneven flow (channeling) and retention of blood flow are likely to occur [11,12,13,14]. Uneven flow and retention generate blood coagulation reaction, and reduce the blood flow channel area, thus increasing the pressure drop. Therefore, the blood flow channel of the newly developed membrane oxygenator is a simple circular shape to suppress uneven flow and retention. This design concept suppresses an uneven flow and retention and makes the blood flow velocity uniform [12,13,14,15,16]. On the other hand, a blood flow channel that increases gas transfer rate by making turbulent blood flow, such as the crossflow, is desired [12,15,16].

Although the quantitative analysis was not sufficient in these experiments, it was possible to clarify the local air and blood retention inside the membrane oxygenator, which has not been understood by the conventional CFD. This is consistent with the results of local blood coagulation inside a membrane oxygenator after clinical use. In addition, the amount of air retention inside the membrane oxygenator is affected by the priming skill of each perfusionist. In clinical practice, it is ideal to completely remove the air inside the membrane oxygenator by the priming operation. However, it is difficult to tell whether the air was actually completely removed. In these experiments as well, since air may not have been removed completely after the priming operation, it was possible that bubble retentions were observed when fluids were perfused (Figure 6 and Figure 7).

Membrane oxygenator dysfunctions in ECMO are caused by excessive pressure drop, due to blood coagulation and thrombus formation, and reduction of gas transfer rate, due to the reduction of effective membrane area, and plasma leakage [1,24,25]. The same emergency supports are performed in ECMO as for the pulmonary dysfunction in CPB. However, it has been reported that, blood coagulation and thrombosis are likely to occur, due to vascular inflammation in patients with COVID-19, and hemofilter is easily clogged in continuous hemodiafiltration (CHDF) treatment [1]. There is concern that the ECMO treatment of patients with COVID-19 will cause a more serious excessive pressure drop. Therefore, our pediatric membrane oxygenator that suppresses high-pressure drop is very useful for ECMO, which is used for longer periods than CPB in cardiovascular surgery.

On the other hand, plasma leakage is likely to occur in ECMO. It has been reported that when hollow fiber membranes of an oxygenator deteriorated, plasma components in the blood leaked into the lumen of hollow fiber (plasma leakage), a liquid of yellow foam leaks from the gas outlet port of an oxygenator during an ECMO treatment for critically ill patients with COVID-19 [1]. At that time, there is concern that COV-19 in plasma may permeate through the pore and diffuse as an aerosol from the gas outlet port, which is still one of the issues in operating ECMO. If the pore diameter of a typical hollow fiber membrane used for membrane oxygenators is on the order of 0.1 µm and the porosity is 40–60% [26], COV-19 with the diameter of 100 nm may permeate through a pore from the outside to inside lumen of hollow fiber membrane. Therefore, first of all, it is necessary to verify the permeability of COV-19, and the appropriate pore size on the outside of the membrane, and anisotropic pore structure design [27,28,29,30] to prevent COV-19 from permeating the anisotropic pore. It would be effective to use a non-porous structure membrane, such as a silicone membrane, to suppress plasma leakage [23,24,25], but a silicone membrane has low gas permeability [26]. Figure 16 shows the observations of the inner lumen and outer surfaces of the capillary membrane of Sample A for reference. The method reported in our previous study [31] was employed to observe the inner lumen and outer surfaces of the capillary membranes using a field emission scanning electron microscope (FE-SEM) (JSM-7610F, Jeol Ltd., Tokyo, Japan) at an accelerating voltage of 1.5 kV and an emission current of 47.2 µA

Polypropylene membrane of the membrane oxygenators is hydrophobic and has very small pores [26], there is sufficient surface tension to prevent plasma infiltration. Hence, these pores are gas-filled, resulting in significantly higher transport rates of oxygen and carbon dioxide than if pores are filled with liquid. However, during ECMO treatment, it is considered that the air in the pores is gradually replaced with plasma (the pores are gradually filled with plasma) over time, resulting in the permeation of plasma into a hollow fiber membrane lumen. Therefore, it is useful to design a pore size and pore structure on the outer side of a membrane such that plasma does not reach the membrane thickness. In the future, it is desired to study the pore structure and materials of the hollow fiber membrane to fulfill the various functions of a membrane oxygenator.

The membrane oxygenator is said to be the “last stronghold” for patients with COVID-19. A membrane oxygenator for artificial lung must be further evolved [24] by taking into consideration of the problems in patients with COVID-19 that we have not experienced before [1].

## 5. Conclusions

This experimental approach identified air and blood retention in the local part of the oxygenator. Our design concept for a membrane oxygenator suppresses air and blood retention and excessive pressure drop. By using this pediatric membrane oxygenator, the abnormally increased pressure drop in the blood flow channel is reduced, and the maximum oxygen transfer rate is raised. It is possible to reduce the incidents and accidents, due to excessive pressure drop that have occurred in clinical practices. Therefore, even in the world’s unprecedented COVID-19 pandemic, this membrane oxygenator for low weight and pediatric patients is promising for ECMO treatment of patients with COVID-19.

## Figures and Tables

**Figure 1 membranes-10-00362-f001:**
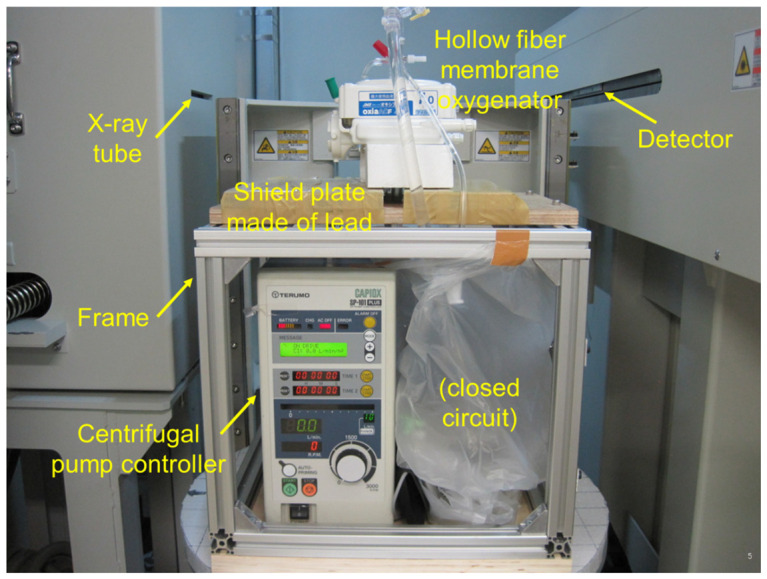
Experimental apparatus for analyzing fluid flow in a device; the non-destructive experimental study using high-power X-ray computed tomography (X-ray CT) [20]. TOSCANER-24500twin (Toshiba Co., Ltd., Tokyo, Japan), installed at the Industrial Technology Center of Wakayama Prefecture.

**Figure 2 membranes-10-00362-f002:**
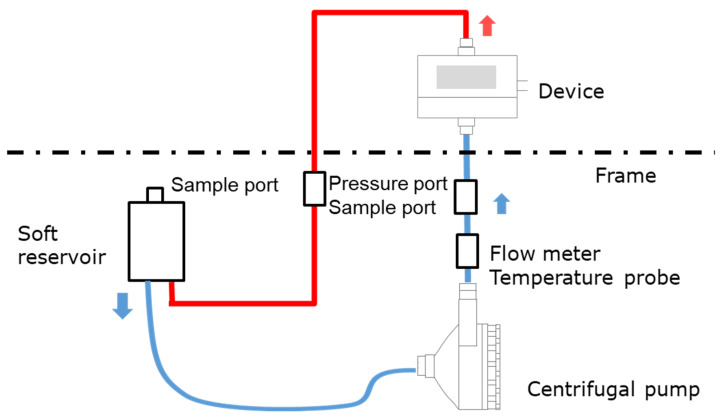
Schematic of an experimental apparatus for analyzing fluid flow in a device; a closed-circuit connected to a soft reservoir.

**Figure 3 membranes-10-00362-f003:**
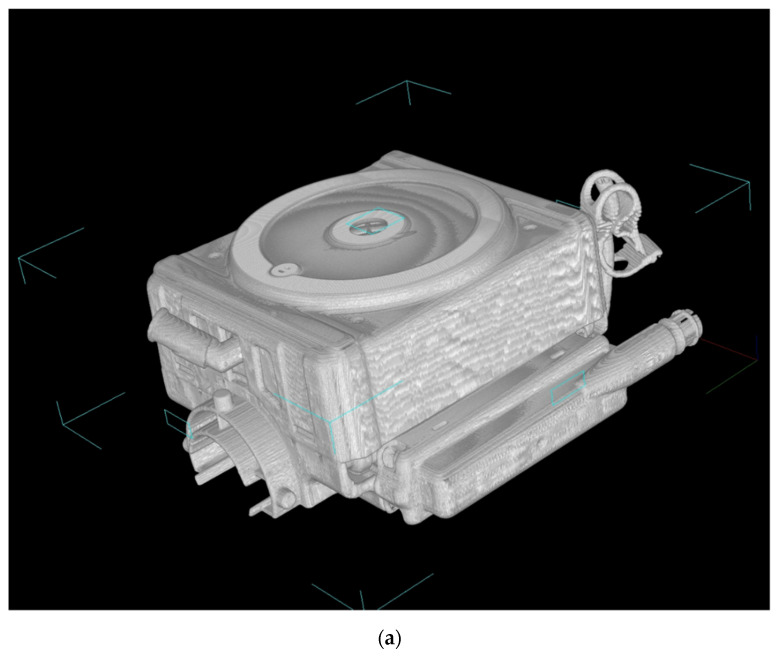
Images of the hollow fiber membrane oxygenator (sample A) by high-power X-ray computed tomography; (**a**) external view, (**b**) vertical cross-sectional view. (**c**) Photographic image of the vertical cross-sectional view. Colored (red, pink, yellow) arrows represent images of blood streamlines. (**b**) (b)-1 Interface between the heat exchanger and blood flow channel. (b)-2 Spacer, 1st layer, the interface between the heat exchanger and spacer. (b)-3 Spacer, 2nd layer. (b)-4 Interface between the heat exchanger and packed hollow fiber membrane (gas exchanger). (b)-5 Packed hollow fiber membrane (gas exchanger). Although the color is dark, the same circular channel as in (b)-4 is confirmed. (b)-6 Interface between the packed hollow fiber membrane layer and the arterial filter. (**c**) (b)-1 Heat exchanger. (c)-2 Interface between the heat exchanger and blood flow channel. (c)-3 Spacer, 1st layer, the interface between the heat exchanger and spacer. (c)-4 Spacer, 2nd layer. (c)-5 Interface between the heat exchanger and packed hollow fiber membrane (gas exchanger). (c)-6 Packed hollow fiber membrane (gas exchanger). (c)-7 Interface between the packed hollow fiber membrane layer and the arterial filter.

**Figure 4 membranes-10-00362-f004:**
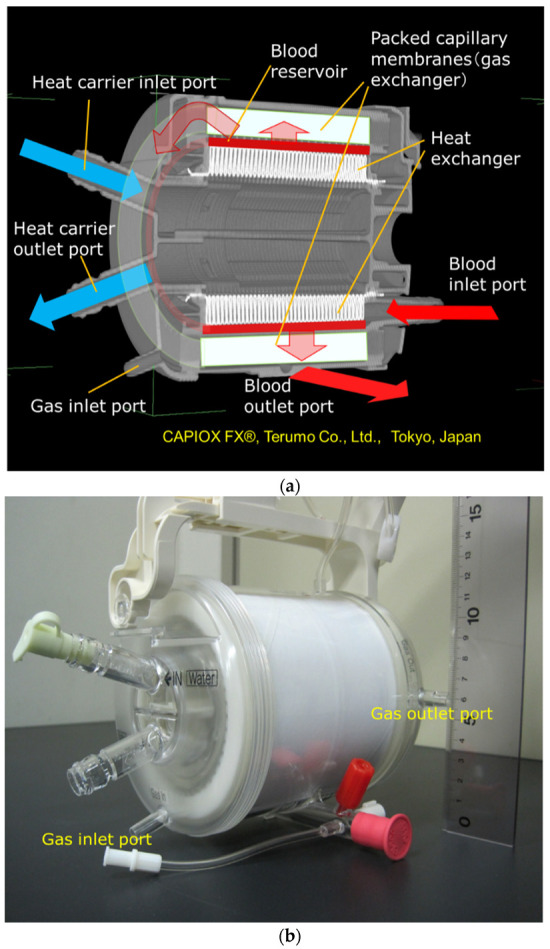
Images of the hollow fiber membrane oxygenator (sample C) by high-power X-ray computed tomography; (**a**) vertical cross-sectional view. (**b**) Photographic image of external view.

**Figure 5 membranes-10-00362-f005:**
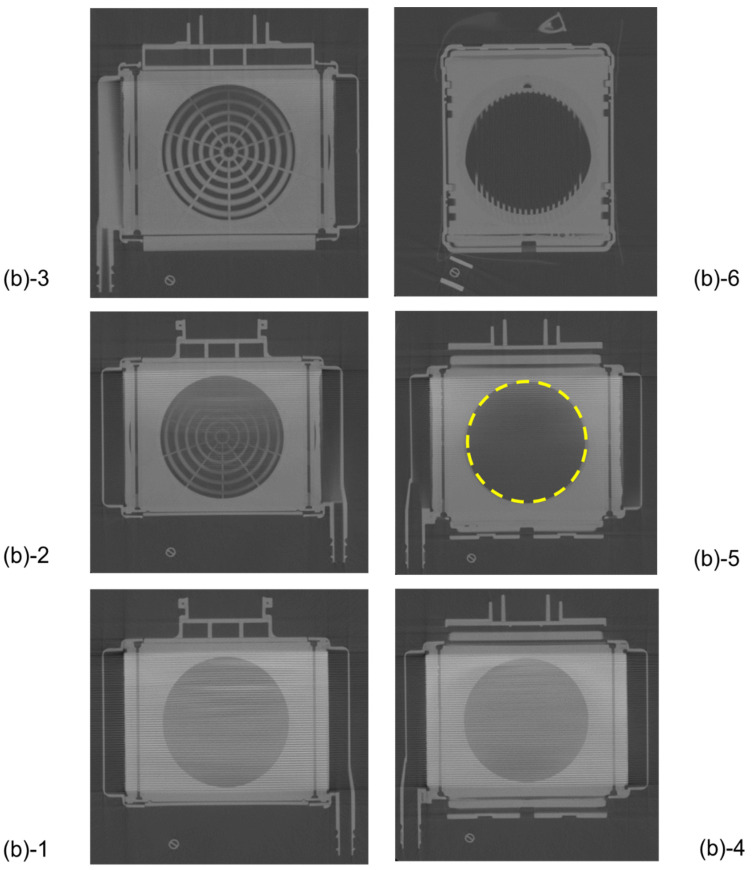
Plane cut view (from top) images of the hollow fiber membrane oxygenator (sample A) by high-power X-ray computed tomography.

**Figure 6 membranes-10-00362-f006:**
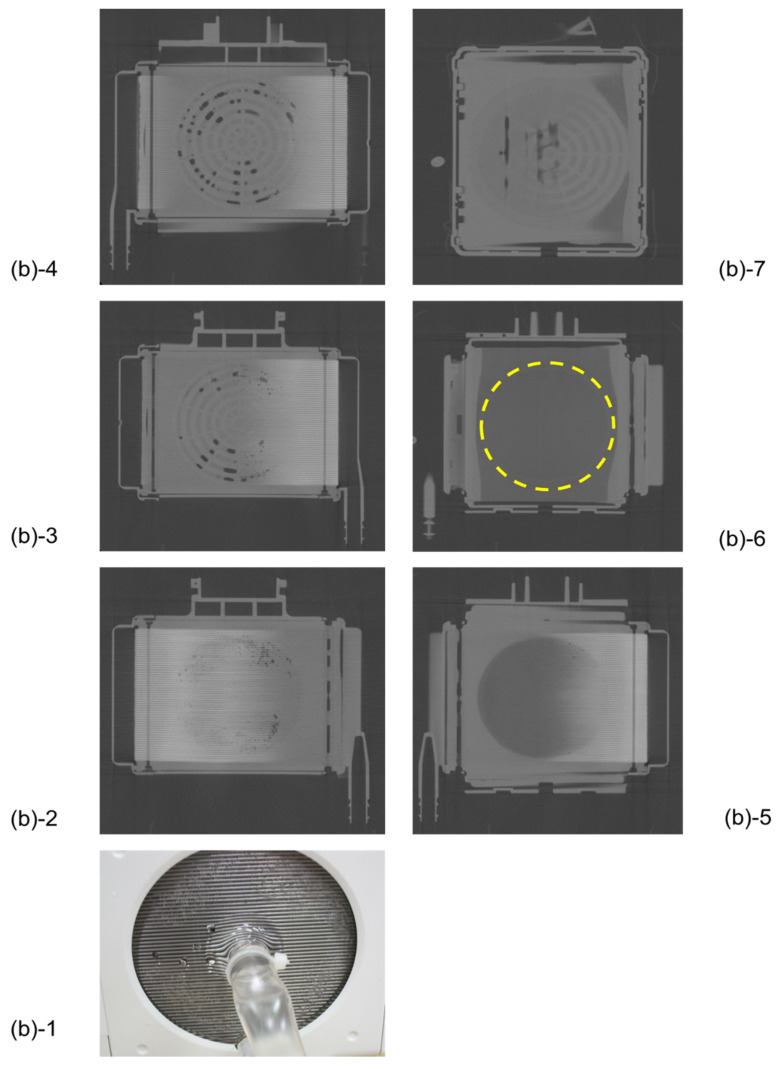
The plane cut view images (from top) of the membrane oxygenator (sample A, oxia^®^ ACF, JMS Co., Ltd., Japan) by the X-ray computed tomography (during RO water perfusion).

**Figure 7 membranes-10-00362-f007:**
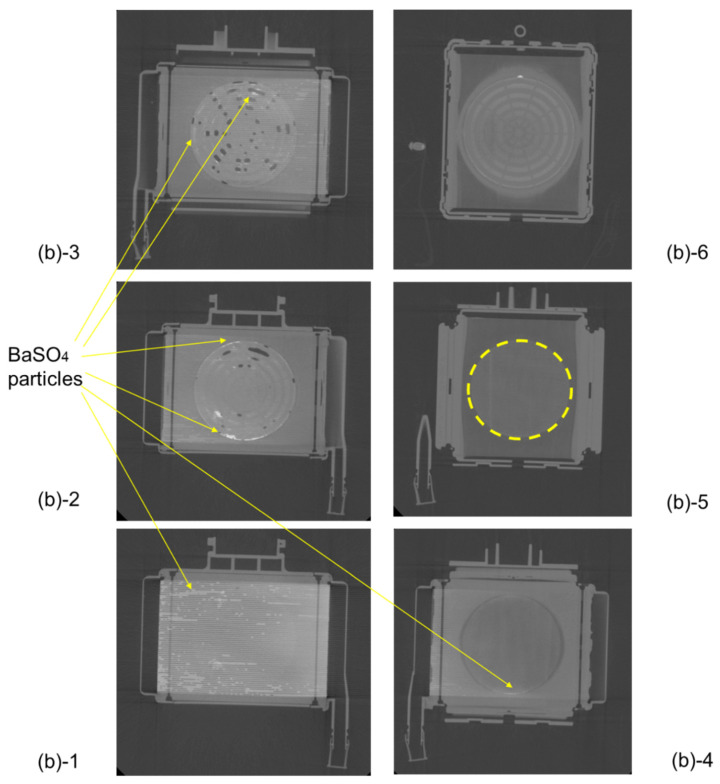
Plane cut view (from top) images of the membrane oxygenator (sample A) by high-power X-ray computed tomography (during RO aqueous solution of 10wt% BaSO_4_ perfusion). Compared with Figure 6, retention of bubbles and X-ray contrast agent particles (BaSO_4_) were more noticeable in the heat exchanger, 1st, and 2nd spacers (Figure 7(**b**)-1~4). There were many air and BaSO_4_ particles at the interface between the spacers and the outer peripheral parts (Figure 7(**b**)-1~4).

**Figure 8 membranes-10-00362-f008:**
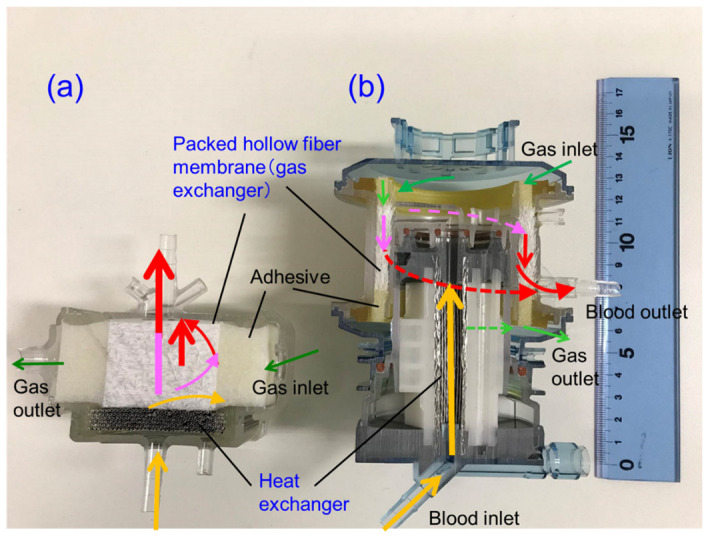
Vertical cross-sectional photographic images of hollow fiber membrane oxygenators, (**a**) newly developed pediatric membrane oxygenator (prototype) and (**b**) commercially available membrane oxygenator (Sample B). Colored (red, pink, yellow) arrows represent images of blood streamlines.

**Figure 9 membranes-10-00362-f009:**
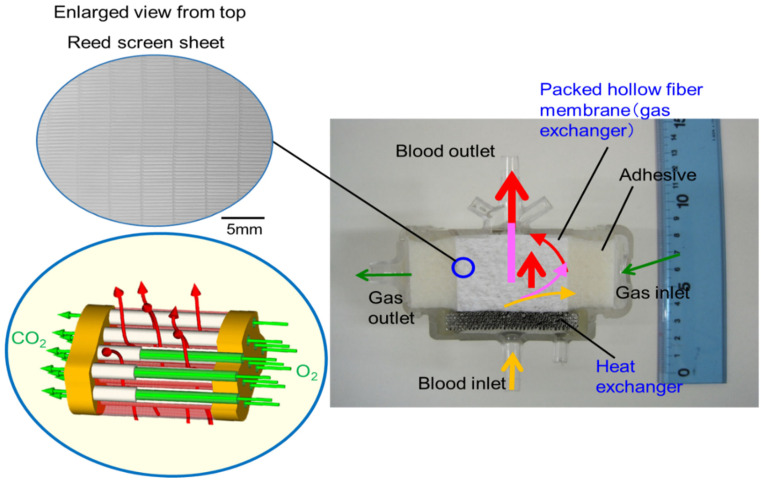
Diagram to explain the geometry of the newly developed membrane oxygenator and the flow of gas and bloodstreams. The reed screen sheets composed of hollow fiber membranes are alternately laminated to attain the multilayer structure of the hollow fiber bundle. Colored (red, pink, yellow) arrows represent images of blood streamlines. The heat carrier is perfused inside of the stainless-steel tube (heat exchanger) from the back of the figure toward the front.

**Figure 10 membranes-10-00362-f010:**
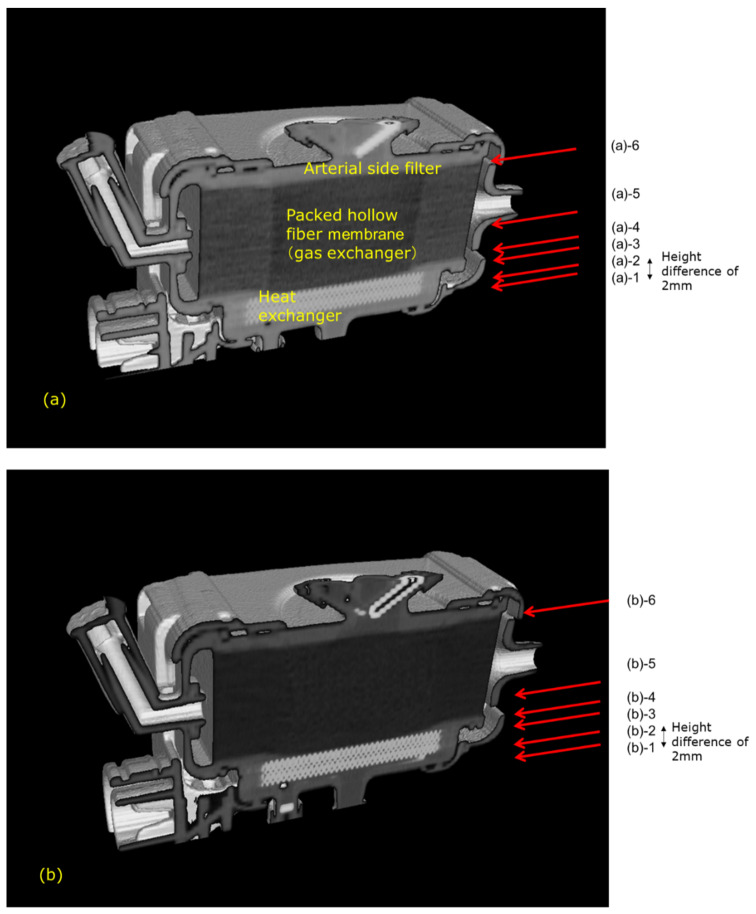
Vertical cross-sectional images of the newly developed hollow fiber membrane oxygenator (prototype) by high-power X-ray computed tomography, (**a**) during RO water perfusion, (**b**) during RO aqueous solution of 10 wt% BaSO_4_ perfusion. (**a**) (a)-1 Heat exchanger. (a)-2 Interface between the heat exchanger and blood flow channel. (a)-3 Interface between the heat exchanger and blood flow channel. (a)-4 Packed hollow fiber membrane (gas exchanger). (a)-5 Packed hollow fiber membrane (gas exchanger) 2mm away from the image in (a)-4. Although the color is darker than that in (a)-4, the same circular channel as in (a)-4 and in (a)-5 is confirmed. (a)-6 Interface between the spacer and arterial side filter. (**b**) (b)-1 BaSO_4_ particles (white) are trapped around outer parts of the blood flow channel. (b)-2 Interface between the heat exchanger and blood flow channel. (b)-3 Interface between the heat exchanger and blood flow channel. (b)-4 Packed hollow fiber membrane (gas exchanger). (b)-5 Packed hollow fiber membrane (gas exchanger) 2 mm away from the image in (b)-4. Although the color is darker than that in (b)-4, the same circular channel as in (b)-4 and in (b)-5 is confirmed. (b)-6 Interface between the spacer and arterial side filter.

**Figure 11 membranes-10-00362-f011:**
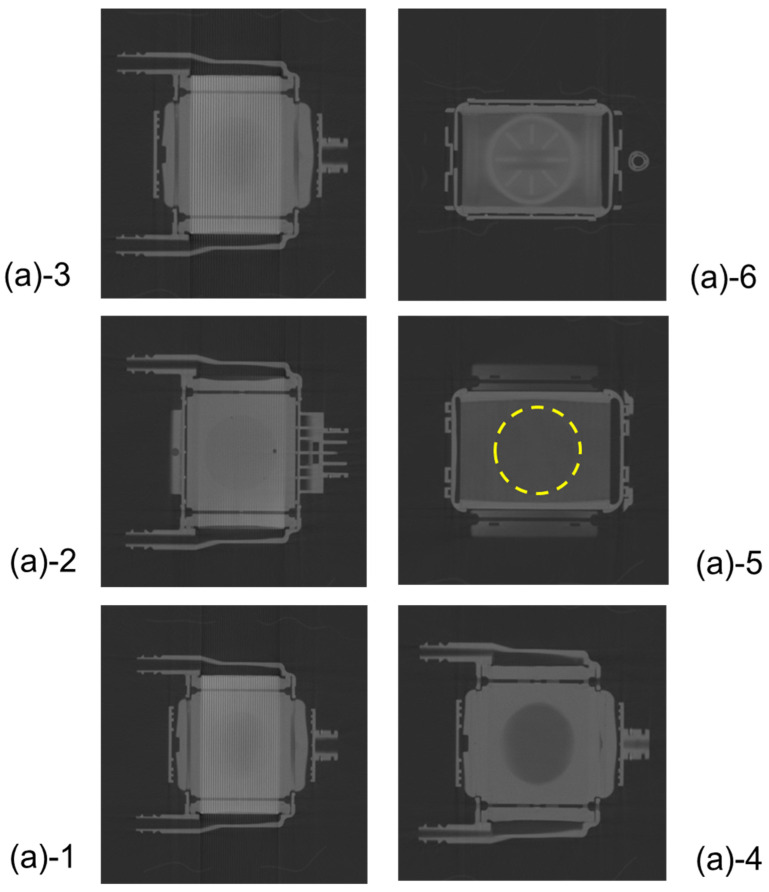
The plane cut view (from top) images of the newly developed membrane oxygenator (prototype) by high-power X-ray computed tomography (during RO water perfusion).

**Figure 12 membranes-10-00362-f012:**
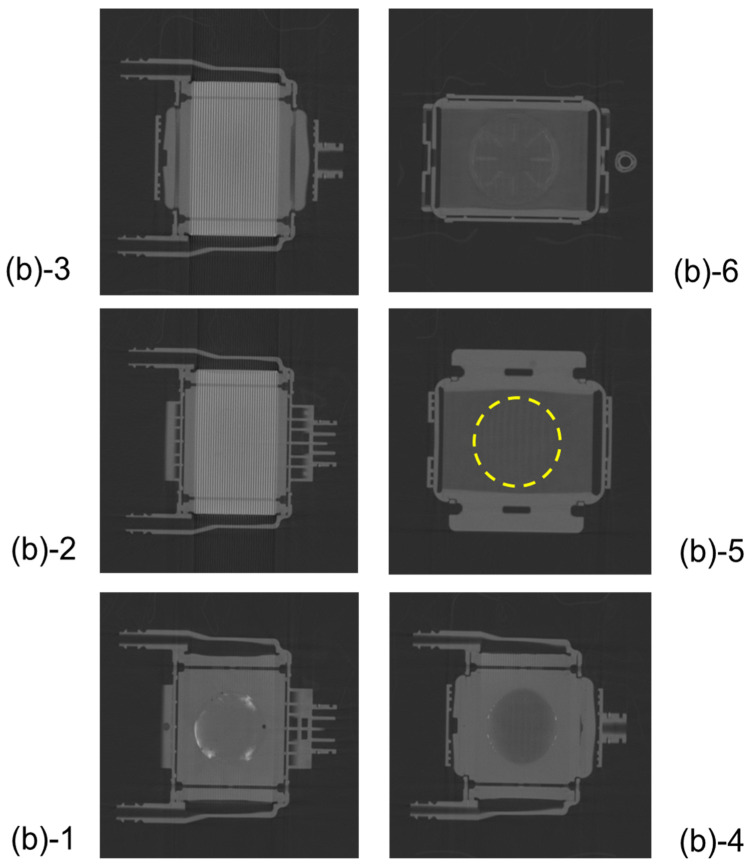
The plane cut view (from top) images of the newly developed membrane oxygenator (prototype) by high-power X-ray computed tomography (during RO aqueous solution of 10 wt% BaSO_4_ perfusion). There is a slight amount of X-ray contrast agent retention at the outermost periphery of the blood flow path in Figure 12(**b**)-1.

**Figure 13 membranes-10-00362-f013:**
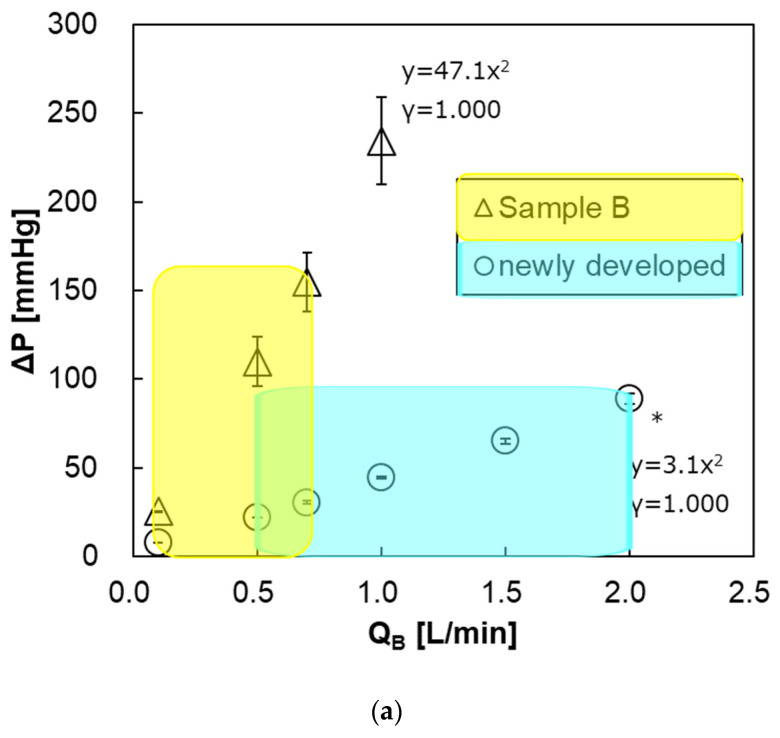
The relationship between pressure drop of the blood flow channel and blood flow rate. * Two-way analysis of variance (two-way ANOVA, *p* < 0.05). Blue and yellow areas represent the recommended ranges of blood flow rate. (**a**) newly developed pediatric membrane oxygenator (prototype) vs. commercially available membrane oxygenator (Sample B). (**b**) newly developed pediatric membrane oxygenator (prototype) vs. commercially available membrane oxygenator (Sample A).

**Figure 14 membranes-10-00362-f014:**
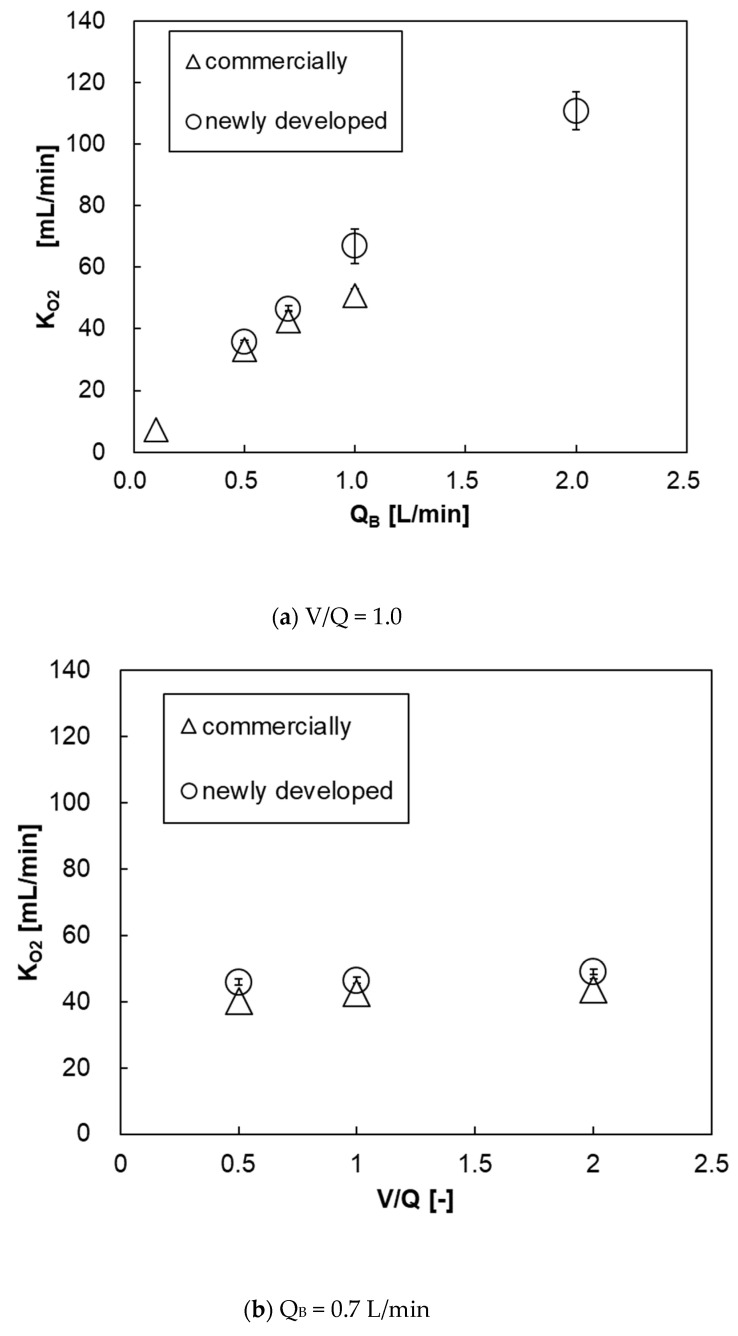
The relationship between oxygen transfer rate and blood flow rate of both pediatric membrane oxygenators, (**a**) V/Q and (**b**) Q_B_ = 0–2 L/min, V/Q = 0.5–2. Two-way analysis of variance (two-way ANOVA, *p* < 0.05).

**Figure 15 membranes-10-00362-f015:**
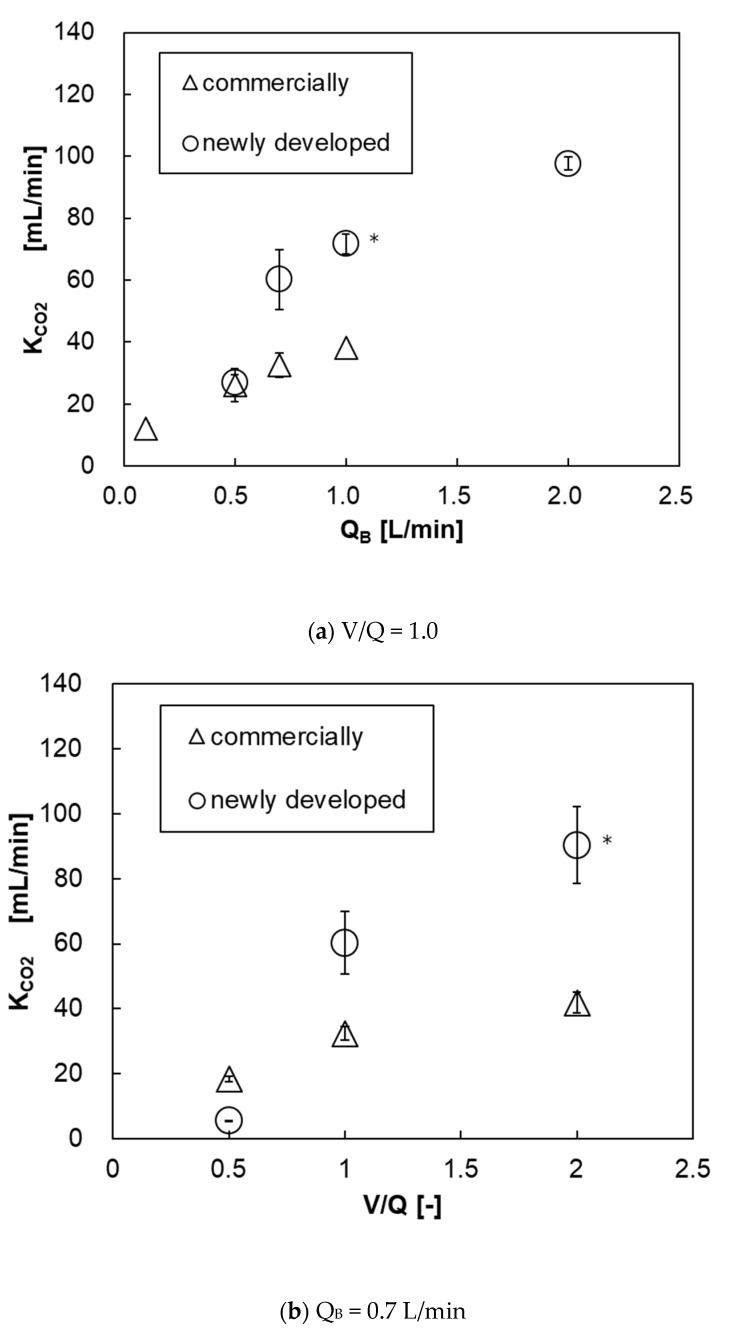
The relationship between carbon dioxide transfer rate and blood flow rate of both pediatric membrane oxygenators, (**a**) V/Q and (**b**) Q_B_ = 0–2 L/min, V/Q = 0.5–2. * Two-way analysis of variance (two-way ANOVA, *p* < 0.05).

**Figure 16 membranes-10-00362-f016:**
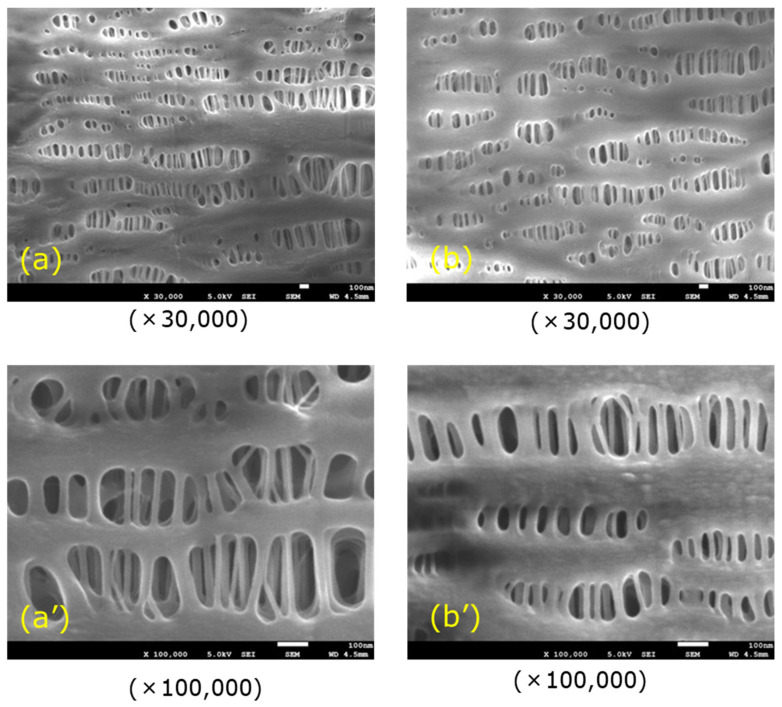
FE-SEM images of the surfaces of the capillary membrane (Sample A), (**a**) inner lumen surface, and (**b**) outer surface.

**Table 1 membranes-10-00362-t001:** Specification of newly developed pediatric membrane oxygenators and other membrane oxygenators.

Sample (Product)	Commercially Available Membrane Oxygenator Sample A	Newly Developed Pediatric Membrane Oxygenator Prototype	Commercially Available Membrane Oxygenator ^1)^ Kids D100 Sample B	Commercially Available Membrane Oxygenator CAPIOX^®^ FX05 Sample C
Manufacturer	JMS Co., Ltd., Japan	―	LivaNova Co., Ltd., Sorin Group, Italy	Terumo Co., Ltd., Japan
Membrane area [m^2^] ^2)^	1.7	0.39	0.22	0.5
Material of hollow fiber membrane Pore structure ^3)^	Polypropylene (PP), asymmetric pore structure	Polypropylene (PP), asymmetric pore structure	Polypropylene (PP)	Polypropylene (PP)
Inner diameter of lumen [µm] (n = 30)	238 ± 5	238 ± 5	233 ± 14	195 ± 8
Membrane thickness [µm] (n = 30)	35 ± 1	35 ± 1	34 ± 1	57 ± 6
Number of hollow fibers [-]	19,500	9000	-	-
Length of hollow fiber [mm]	90	45	-	-
Air permeability ^4)^ [sec]	35.1 ± 1.4	33.8 ± 0.4	-	-
Antithromboge-nic material coating for blood flow channel	Poly(2-methacryloyloxyethyl phosphoryl choline) (PMPC)	Poly(2-methacryloyloxyethyl phosphoryl choline) (PMPC)	poly(2-methacryloyloxyethyl phosphoryl choline) (PMPC)	poly(hydoroxyethylmetharylate) (PHEMA)
Priming volume [mL]	245	37	31	43
Range of blood flow rate (Max Q_B_) [L/min]	1.0–7.0 (7.0)	0.5–2.0 (2.0)	0.1–0.7 (0.7)	−1.5 (1.5)
Calculated retention time [s] ^5)^	2.1	1.1	2.6	1.7

^1)^ The commercially available pediatric membrane oxygenator, which is currently the smallest in the world. ^2)^ Membrane area = outer diameter of hollow fiber lumen × π × length of hollow fiber × number of hollow fibers. ^3)^ Tortuous pore diameter is not measured. ^4)^ The gas permeability of the membranes was measured by the Gurley method using an air resistance tester defined in ISO5636-5 [19]. ^5)^ Calculated retention time = Priming volume/Maximum blood flow. The smaller the value, the larger the gas permeability.

**Table 2 membranes-10-00362-t002:** Maximum oxygen and carbon dioxide transfer rates and pressure drop of membrane oxygenators.

Sample (Product)	Commercially Available Membrane Oxygenator Sample A	Newly Developed Pediatric Membrane Oxygenator Prototype	Commercially Available MEMBRANE Oxygenator Kids D100 Sample B
Maximum oxygen transfer rate [mL/min, STP]	433 (Max Q_B_ = 7 L/min)	116 (Max Q_B_ = 2 L/min)	43 (Max Q_B_ = 0.7 L/min)
Maximum carbon dioxide transfer rate [mL/min, STP] (V/Q = 1.0)	355	98	38
Pressure drop of the blood flow channel [mmHg]	106 (Max Q_B_ = 7 L/min)	89 (Max Q_B_ = 2 L/min)	155 (Max Q_B_ = 0.7 L/min)

Pressure drop of the blood flow channel (packed capillary membranes and heat exchanger).

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
