# Peer review of "Newly Developed Pediatric Membrane Oxygenator that Suppresses Excessive Pressure Drop in Cardiopulmonary Bypass and Extracorporeal Membrane Oxygenation (ECMO)"

_membranes, 2020, doi:10.3390/membranes10110362_

Round 1
Reviewer 1 Report
The authors present a way to improve the current membrane oxygenator for pediatric uses to reduce the excessive pressure drop within the exchanger caused by clogging or coagulations related to blood or air. They used an X-ray tracer Barium sulfate that can be directly visualized to probe potential design space available for the improvement and designed a new hollow fiber packing with an enhanced mass transfer coefficient. The current study is fascinating and matches with the urgent need in the current pandemic situation. I only have one question and one minor suggestion for the claim made in this study.
While many of the cloggings of solutes on the membrane are related to non-specific interactions, using the inorganic salts with DI water is not likely to accurately represent the actual clogging site. Can the authors answer why the tracer solution was not combined with bovine blood?
Furthermore, the authors mention that plasma leaking is a critical challenge during actual blood oxygen usage, and the virus may permeate through. It would be more helpful if the membranes' SEM images are included to show the actual pore size, although the PP used in this current study is like to fall in the microfiltration range.
Reviewer 2 Report
Why did the authors use an aqueous suspension of barium sulfate for their research. Typically an albumin solution (BSA) is used to simulate blood. Blood has distinctly different rheological properties than water. Here it is necessary to justify such a choice.
Platelet fibrin thrombus and coagulation is not only caused by heterogeneous blood flow. There are many reasons for this, eg surface roughness, sharp edges, materials used. This paper does not mention other causes of blood coagulation in the oxygenator.
It is not clear from the text whether the tests were carried out on one copy of the prototype or on several copies. Even the very interesting results obtained from testing one specimen are of limited value. Such information should be clearly stated.
The conclusions from the research seem to be accurate and interesting. But in view of the previous comment, they may be of limited value.
